# Absolute energy level positions in tin- and lead-based halide perovskites

Shuxia Tao [1], Ines Schmidt[2], Geert Brocks[1,3], Junke Jiang [1], Ionut Tranca[4], Klaus Meerholz[2] & Selina Olthof[2]

Metal halide perovskites are promising materials for future optoelectronic applications. One intriguing property, important for many applications, is the tunability of the band gap via compositional engineering. While experimental reports on changes in absorption or photo-luminescence show rather good agreement for different compounds, the physical origins of these changes, namely the variations in valence and conduction band positions, are not well characterized. Here, we determine ionization energy and electron affinity values of all primary tin- and lead-based perovskites using photoelectron spectroscopy data, supported by first-principles calculations and a tight-binding analysis. We demonstrate energy level variations are primarily determined by the relative positions of the atomic energy levels of metal cations and halide anions and secondarily influenced by the cation-anion interaction strength. These results mark a significant step towards understanding the electronic structure of this material class and provides the basis for rational design rules regarding the energetics in perovskite optoelectronics.

[1] Center for Computational Energy Research, Department of Applied Physics, Eindhoven University of Technology, P.O. Box 513, 5600MB Eindhoven, The Netherlands. [2] Department of Chemistry, University of Cologne, Luxemburger Straße 116, Cologne 50939, Germany. [3] Computational Materials Science, Faculty of Science and Technology and MESA+ Institute for Nanotechnology, University of Twente, P.O. Box 217, 7500 AE Enschede, The Netherlands. [4] Energy Technology, Department of Mechanical Engineering, Eindhoven University of Technology, 5600 MB Eindhoven, The Netherlands. Correspondence and requests for materials should be addressed to S.T. (email: S.X.Tao@Tue.nl) or to S.O. (email: solthof@uni-koel.de)

Metal halide perovskites are solution-processable semi-conducting materials with a general formula $AMX_3$, where A are monovalent cations ($Cs^+$, $MA^+ = (CH_3NH_3)^+$, or $FA^+ = (CH(NH_2)_2^+)$), M are metal cations ($Pb^{2+}$ or $Sn^{2+}$), and X are halide anions ($I^-$ or $Br^-$ or $Cl^-$). This material class has received enormous attention in the scientific community recently due to breakthroughs in perovskite-based optoelectronic applications, mainly in photovoltaics[1–5], but also in photodetectors[6], light emission[7–9], and lasing[10]. Intriguingly, by exchanging or mixing different A, M, and/or X ions, it is possible to tune the optical gap of these semiconductors, which is exploited, e.g., to optimize the overlap with the solar spectrum in tandem solar cells[11], or to tune the emission wavelength of LEDs[12]. These changes in band gap are well characterized for a large number of primary $AMX_3$ compounds, as well as for more complex perovskite mixtures, in experimental[13–15] as well as computational[16–25] studies. However, two fundamental questions have not been resolved yet: (i) what is the underlying physical origin of the changes in the band gaps and (ii) how do the absolute positions of the valence band maximum (VBM) and conduction band minimum (CBM) change as a function of the composition of the perovskites? The answers to these questions are not only fundamentally highly interesting, but are also needed to develop strategies for tailoring desired optoelectronic properties and to optimally match perovskite energy levels to contacts and extraction layers for efficient charge transport through a device[26–30].

The challenges in answering these questions originate from the complex interplay of a few subtle yet correlated factors when combining different A, M, and X, such as the type and the size of ions, the crystal structure, and the degree of distortion with respect to the ideal perovskite structure[18,23]. Experimental studies, mainly based on photoelectron spectroscopy, have been limited to a small subset of $ABX_3$ compounds and suffer from significant variations in reported energy level values, which are due to variations brought on by preparation conditions[31–33], as well as by different data evaluation protocols[34]. Unlike in all-organic semiconductors, it is non-trivial to determine the energy onsets of the valence and conduction bands in metal halide perovskites experimentally using direct and inverse photoelectron spectroscopy due to a low density of states (DOS) at the band edges[35]. To reliably learn something about energy level trends in these systems, comparative studies are needed for subsets of compounds; these are however scarce and mostly limited to halide variations in $MAPbX_3$ perovskites, see e.g. refs. [14,35–37]. Computational studies are also insightful for identifying trends in band gaps[16,18–20,23], key characteristics of DOSs[20–23], and band structures[24,25]. However, predictions of the absolute energy levels and their trends are challenging to make, due to intrinsic approximations in the methods, variations in the choice of structural models[17], and the need to model the crystal terminations (i.e. the surfaces).

Here, we provide answers to these open questions by determining the absolute energy levels of the complete set of primary Sn and Pb perovskites using optimized material fabrication and consistent data evaluation procedures. The energy levels of these compounds are determined by combining UV and inverse photoelectron spectroscopy (UPS and IPES), as well as absorption measurements. We carefully determine the Ionization Energy (IE) and Electron Affinity (EA) by comparing the experimental and density-functional-theory (DFT) calculated DOSs. Incorporating these values into an intuitive tight-binding model, we are able to give a clear analysis of all trends in IEs and EAs. This study therefore provides a fundamental understanding of the evolution of key electronic energy levels of metal halide perovskites and opens up the possibility for rational materials design for efficient perovskite optoelectronic devices.

## Results

**Photoelectron spectroscopy measurements.** Figure 1 shows UPS and IPES measurements on representative films of all 18 tin- and lead-based $AMX_3$ perovskites. To increase the comparability, all spectra have been shifted along the x-axis in such a way that the high energy cutoff position, marked by line A, is located at the excitation energy of 21.22 eV. This way, the positions of $E = 0$ eV corresponds to the vacuum level and the onset positions of the valence band and conduction band directly match the IE and EA values. These VBM and CBM positions are indicated by black vertical markers and are extracted by correlating the measured spectra with DOSs obtained from first-principles calculations as elaborated below.

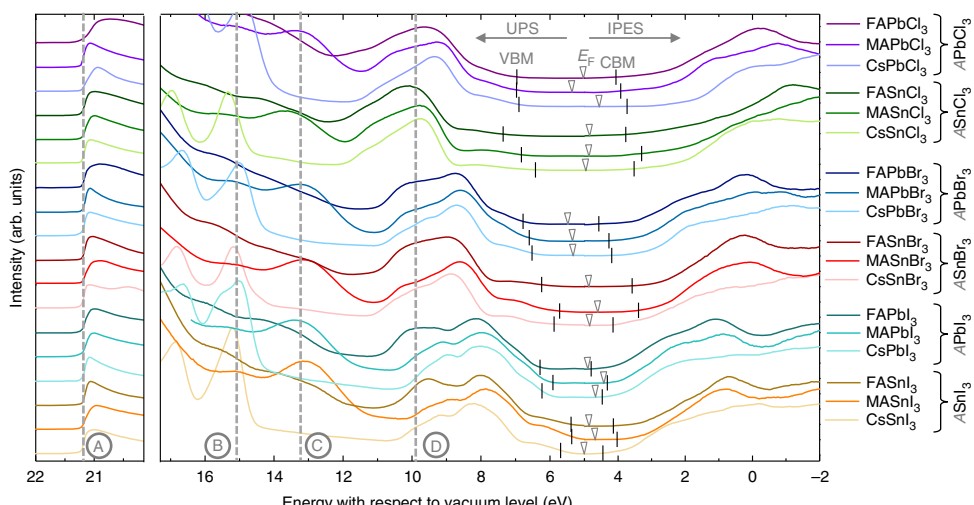

**Fig. 1** Representative UPS and IPES spectra of all 18 metal halide perovskite systems. For better comparability, the curves are offset vertically and the high energy cutoffs are aligned at the excitation energy of 21.22 eV, marked by line A. Lines B, C, and D indicate characteristic features in the DOS, corresponding to the position of Cs, MA, and FA related states, respectively. The extracted positions of VBM and CBM are given by black vertical markers, and the Fermi level positions are marked by triangles. The procedure for evaluation of the exact positions of the VBM and CBM is elaborated in Fig. 2. Note that for IPES measurements only the smoothed data trend is shown (raw data are included in the Supplementary Fig. 6)

The three additional vertical lines in Fig. 1 mark the positions of the Cs $5p_{3/2}$ semicore level (line B at 15.1 eV), a MA molecular level (line C at 13.1 eV) and a FA molecular level (line D at 9.9 eV). Intriguingly, in our experiments we found that these features consistently appear at almost the same positions (within ±0.2 eV) with respect to the vacuum level, independent on the perovskite structure. This finding is further supported by our DFT-derived semi-core levels of Cs $s$ and $p$ states in CsMX₃ perovskites (Supplementary Table 1), where merely minor energetic differences are present (within ±0.1 eV); only for CsSnCl₃, a somewhat larger deviation of ~0.3 eV was found. These key features have been used throughout the experiments as valuable indicators for evaluating proper material preparation procedures. An off-stoichiometry composition usually unintentionally changes the IE of the film[32], which is then also evident by shifts in these atomic and molecular levels. It should be noted that we have extensively optimized the preparation procedure of each compound and have further evaluated the quality of our samples with respect to elemental composition, oxidation state, crystal structure, and morphology; the corresponding measurements can be found in the Supplementary Notes 1–18.

As already indicated above, it is challenging to reliably determine the onset of the bands directly from the experimentally measured DOSs as presented in Fig. 1. For example, the imperfect surface or defects in the film can introduce additional states at the band edges while an insufficient coverage can lead to substrate features appearing in the measured DOS. Furthermore, the typically low DOS of perovskites at the onsets can be missed with common fitting procedures. Such issues can be circumvented to a certain degree by fitting the experimental DOSs at the band edges to the corresponding DFT-calculated DOSs as already put forward by Endres et al.[35]. We employ a similar approach in this work, however instead of focusing on band onsets, we align key features of the experimental and DFT DOSs. Fitting over the full DOS region has the advantage that the results are independent of experimental broadening. Furthermore, possible gap states at the band edges are not factored in.

The fitting procedure we employ, is shown in Fig. 2, using the materials FAPbI₃ and FASnI₃ as examples. Data sets concerning the remaining 16 materials can be found in the Supplementary Fig. 6. First, the DFT DOS is calculated (Fig. 2a, e) and corrected such, that the band gap is equal to the experimentally measured optical gap (UV-vis measurements in the Supplementary Notes 1–18). In some cases the spectra were stretched by a few percent, values are listed in the Supplementary Table 4. Next, each point of the DOS is broadened by a Gaussian function such as to match the experimental resolution, resulting in the data sets shown in the subfigures b and f. Further details can be found in the discussion of Supplementary Fig. 5. To be able to match theory and experiment, these broadened DFT spectra are then fitted to a linear combination of Gaussian peaks; these are chosen such that they allow for a robust fit and are able to consistently describe the various perovskite compositions. The peaks roughly correspond to the partial DOS features, as can be seen when comparing them to the features in c and g. Finally, the same sets of Gaussian peaks are used to fit the experimental DOS, shown in Fig. 2d, h; by this, the experimental and DFT DOSs can be aligned as indicated by the colored regions in Fig. 2. The valence and conduction band onsets can be obtained from the VBM and CBM in the original (i.e., not broadened) DFT spectra, as indicated by the dashed lines.

Our procedure proves to be very robust for the VB region and therefore the position of the VBM can be accurately determined. However, in the CB region the agreement between the calculated and experimental DOSs is not as good, which makes it difficult to align them. Currently, it is unclear where the difference comes

from, as both the calculations and experiments match previously reported calculated[35,38–40] and measured spectra[35–37,41,42] well; in addition, the sample-to-sample variation is small (Supplementary Fig. 6). We suggest that the inconsistency between the DFT and IPES-derived DOSs may have to do with significant differences in measurement cross section. Due to the uncertainties, we aligned DFT and IPES-derived DOSs using only the first two CB features. This is more error prone than a fit over a wider region, and additional constraints are needed to make the fit more robust. Therefore we included the constraint that the electronic gap has to be close to the optical gap. Differences in electronic and optical band gaps of common three-dimensional metal halide perovskites are typically small (tens of meV)[43–46] and are in the same order of magnitude as standard sample-to-sample variations.

Using the above described fitting routine, we determine the absolute positions of VBM and CBM for all Pb and Sn perovskites, as indicated by the black vertical markers in Fig. 1. The extracted values for the IE and the EA are listed in Table 1, together with the optical gap obtained from the UV-vis measurements (Supplementary Notes 1–18). For each material we average these values over three separate samples, the error bars in Table 1 correspond to the sample-to-sample variation; the individual UPS/IPES spectra and fits are shown in the Supplementary Fig. 6.

Most of these materials have not been reported before, so we cannot in general discuss comparability of the values found here to previous reports. Even for the more extensively studied systems, published values usually scatter quite significantly due to issues in film preparation and data evaluation, as already mentioned in the introduction. For example, for MAPbI₃ values between 5.1 eV and 6.65 eV have been published[34] and for MASnI₃ a similarly broad distribution of 4.73–5.47 eV is found[47–50]. This emphasizes once more the need for a consistent study with an unambiguous data evaluation process. Looking at Table 1, large differences in both energy level positions and band gaps are found between the different compounds. These trends will be discussed next based on a chemical bonding analysis and a tight-binding model.

**The chemical bonding in AMX₃ perovskites.** To rationalize the observed energy differences, we probe the contributions of the different atoms to the calculated DOS (called the partial DOS here). The example of CsPbI₃ is shown in Fig. 3a, where for simplicity the position of VBM is set to zero. Consistent with results reported in the literature[21–25], the states at the CB and VB band edges are dominated by Pb and I contributions, whereas Cs-related states are found at much lower and at much higher energies. The sharp feature around −8 eV in the DOS corresponds to a quasi-atomic state of the Cs⁺ ion, for instance.

Next, we analyze the electronic structure using a Crystal Orbital Hamiltonian Population (COHP) analysis. A COHP analysis gives the covalent bonding and anti-bonding character of the DOS at each energy, and indicates which atomic orbitals are involved. Figure 3a, b show a direct comparison between the DOS and the COHP of CsPbI₃ in the energy range between −13 eV and +5 eV. Cs character is almost absent in the orbital-resolved COHP, implying that Cs does not participate in covalent bonding; clearly, Pb and I contributions dominate. This pattern is observed in all AMX₃ compounds studied here. We therefore concentrate on the interactions of metal (M) cation and halide (X) anion, excluding those involving A site cations. Four pairs of bands of bonding/anti-bonding states can be identified in Fig. 3b (for the case of CsPbI₃), which result from the hybridization of the $s$ and the $p$ orbitals of the Pb and I atoms. In particular, the

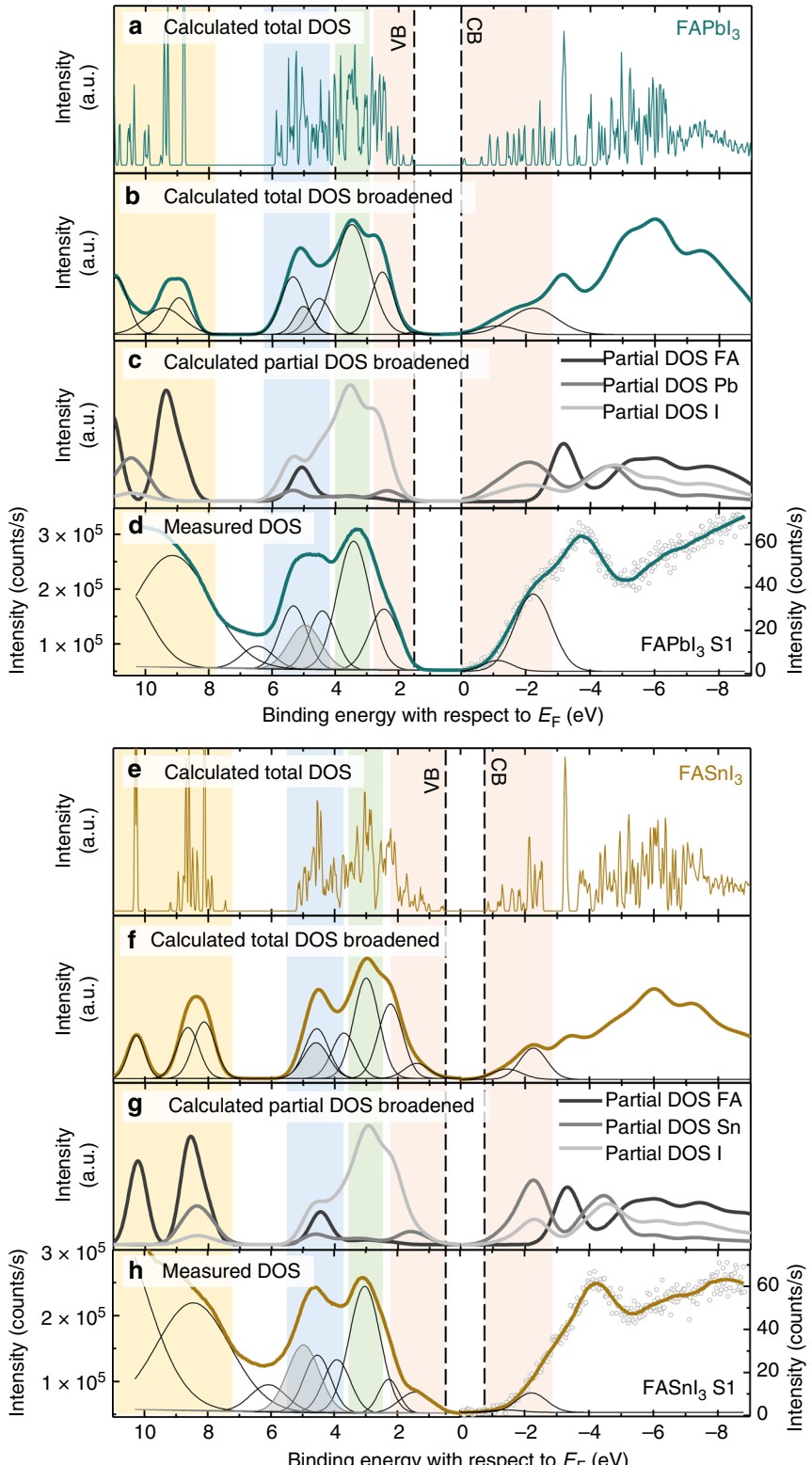

**Fig. 2** Comparisons between measured and DFT-calculated densities of states. **a** Calculated DOS of FAPbI$_3$. **b** Same DOS uniformly broadened by a Gaussian function and subsequently this broadened DOS is fitted with Gaussian peaks. **c** Similarly broadened DOS, but showing the partial DOS contributions of the perovskites constituents. **d** Experimental DOS of FAPbI$_3$, measured by UPS and IPES, fitted with a similar set of Gaussian peaks, so that the DFT and experimental spectra can be aligned. Subfigures **e** to **h** show the equivalent plots for FASnI$_3$. The different colored vertical bars act as guide for the eye to show how the spectra are lined up while the dashed lines show the onset of the VB and CB in the unbroadened DFT data; the shaded gray peak corresponds to FA related features. Note that the x-axes are plotted with respect to the Fermi level and the DFT spectra have been shifted to match their Fermi energies of the measured data sets. Further, the DFT data are plotted (in accordance with the measurement) using positive energies for VB and negative energies for CB bands; in Figs. 3 and 4 we use opposite signs for the energies, in agreement with standard DFT convention

**Table 1 List of extracted IEs, EAs, and optical band gaps $E_{g,opt}$**

| | Pb based | | | Sn based | | | |
|---|---|---|---|---|---|---|---|
| | I | Br | Cl | I | Br | Cl | |
| $E_{g,opt}$ | 1.72 ± 0.01 | 2.31 ± 0.1 | 2.99 ± 0.02 | 1.25 ± 0.02 | 1.81 ± 0.1 | 2.88 ± 0.05 | Cs |
| IE | 6.25 ± 0.1 | 6.53 ± 0.05 | 6.80 ± 0.1 | 5.69 ± 0.1 | 5.82 ± 0.1 | 6.44 ± 0.05 | |
| EA | 4.47 ± 0.1 | 4.17 ± 0.1 | 3.77 ± 0.1 | 4.38 ± 0.1 | 4.07 ± 0.1 | 3.47 ± 0.1 | |
| $E_{g,opt}$ | 1.59 | 2.30 ± 0.02 | 3.04 ± 0.01 | 1.24 ± 0.02 | 2.13 ± 0.02 | 3.50 ± 0.08 | MA |
| IE | 5.93 ± 0.05 | 6.60 ± 0.05 | 6.92 ± 0.1 | 5.39 ± 0.05 | 5.67 ± 0.05 | 6.85 ± 0.15 | |
| EA | 4.36 ± 0.1 | 4.25 ± 0.1 | 3.77 ± 0.15 | 4.07 ± 0.1 | 3.42 ± 0.1 | 3.36 ± 0.15 | |
| $E_{g,opt}$ | 1.51 ± 0.02 | 2.25 ± 0.02 | 3.02 ± 0.05 | 1.24 ± 0.1 | 2.63 ± 0.1 | 3.55 ± 0.05 | FA |
| IE | 6.24 ± 0.1 | 6.7 ± 0.1 | 6.94 ± 0.05 | 5.34 ± 0.1 | 6.23 ± 0.05 | 7.33 ± 0.1 | |
| EA | 4.74 ± 0.15 | 4.51 ± 0.1 | 3.98 ± 0.1 | 4.12 ± 0.1 | 3.6 ± 0.1 | 3.83 ± 0.1 | |

The values are extracted from data given in Fig. 1, as well as data from the Supplementary Fig. 6 and Supplementary Notes 1–18. All values, which are given in eV, are averaged over 3 samples each; the error bars represent the spread over these samples

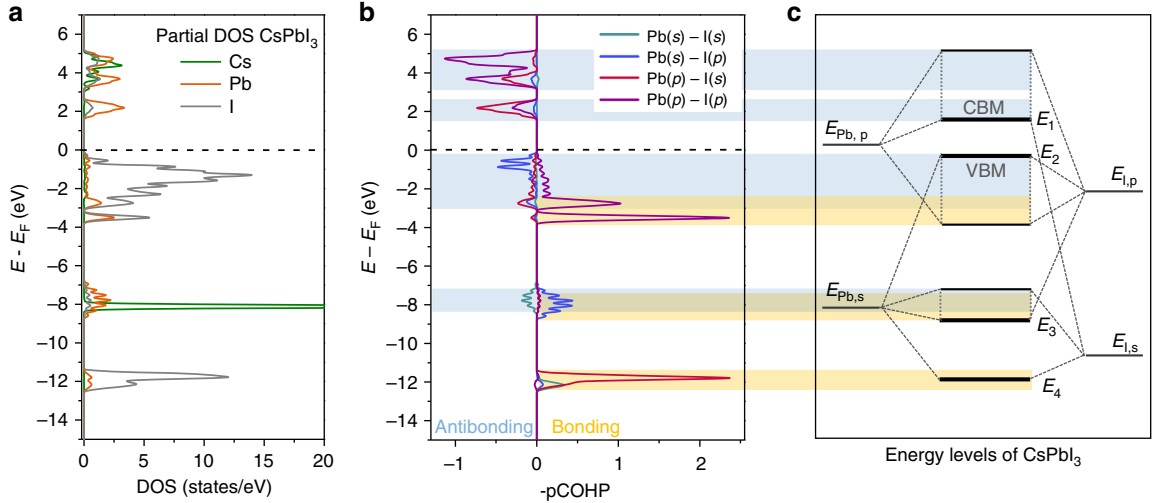

**Fig. 3** Orbital contributions to the energy bands of CsPbI₃. **a** CsPbI₃ DOS projected on the Cs, Pb, and I atoms (partial DOS). **b** Orbital-resolved COHP; positive (negative) sign indicates bonding (anti-bonding) character. The line colors indicates which atomic orbitals are involved in the bonds (anti-bonds). **c** Schematic energy level diagram extracted from the COHP analysis; the energy levels $E_1$ to $E_4$ which are relevant for the tight-binding analysis are marked. Bonding interactions are shaded in yellow, and anti-bonding interactions in blue

top of the VB can be identified as a Pb,$s$/I,$p$ anti-bonding state, whereas the bottom of the CB is a Pb,$p$/I,$s$ anti-bonding state. From the COHP we construct the simplified energy level diagram shown in Fig. 3c. Here all bands are interpreted in terms of hybridization of the atomic $s$ and $p$ states of Pb and I. In the tight-binding analysis, to be described below, we represent the energy levels of those atomic states by $E_{Pb,s}$, $E_{Pb,p}$, $E_{I,s}$, and $E_{I,p}$. The energies of the CBM and the VBM are indicated by $E_1$ and $E_2$ in Fig. 3c. As remarked above, both of these levels correspond to anti-bonding states, while their bonding partners can be found at energies $E_4$ and $E_3$, respectively. These four states $E_{1-4}$ play a central role in our tight-binding analysis.

**Tight-binding analysis**. Our tight-binding analysis focuses on the VBM and the CBM. The analysis becomes more straightforward if one concentrates on cubic symmetry and uses the symmetry analysis presented by Boyer-Richard et al.[19]. For cubic perovskites (space group P$m$-3$m$) the VBM and CBM are situated at the R-point of the Brillouin zone, and one can restrict a tight-binding analysis to states at the R-point.

We apply a nearest neighbor tight-binding model with six parameters: the four effective atomic $s$ and $p$ energy levels of the

M cation and the X anion, see Fig. 3c, and two hybridization strengths, between the M,$s$ and X,$p$ orbitals, and between the M,$p$ and X,$s$ orbitals, respectively. Interactions between M,$s$ and X,$s$ orbitals, and between M,$p$ and X,$p$ are symmetry forbidden at the R-point of a cubic perovskite, so we do not have to consider the corresponding hybridization strengths[19]. The energy levels of the halide ions, $E_{X,s}$, $E_{X,p}$, can be obtained from the DFT-calculated level spectra at the Γ-point or the R-point, by identifying halide states that are non-bonding in cubic perovskites. The remaining four parameters can then be calculated from the energy levels $E_{1-4}$ in the DFT-calculated level spectrum at the R-point. For details, see the Methods Section. The dominant effect of spin-orbit-coupling (SOC) on the electronic structure stems from the SOC-induced $p$-level splitting on the M cation. We include this as an atomic parameter $\Delta_{SOC}$, where we use $\Delta_{SOC}$ of 1.65 eV and 0.60 eV for Pb and Sn, respectively[23,51].

A graphical representation of the results of the tight-binding analysis at the R-point is given in Fig. 4a, b; the corresponding values of all relevant energy levels can be found in the Supplementary Table 3. In the following we use these results to analyze the trends in the VBM and CBM in case the halide anions or the metal cations are exchanged. We also discuss the influence of the structural variations in volume and distortions on

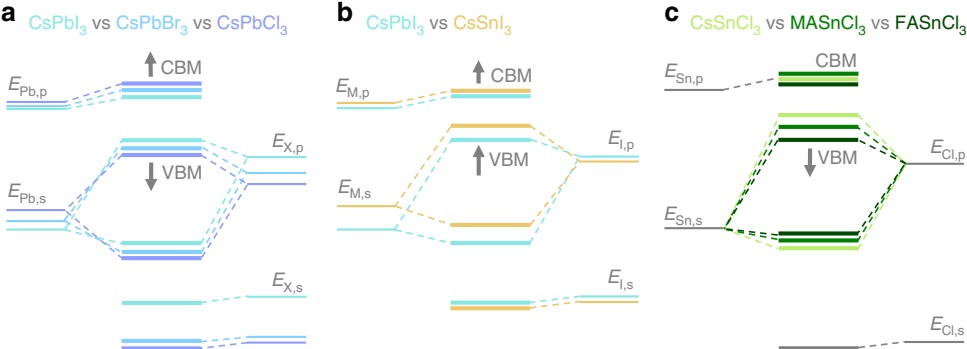

**Fig. 4** Schematic energy levels in AMX$_3$ perovskites. **a**, **b** represent trends in changing the halide anions and the metal cations, respectively, as identified from the tight-binding analysis. **c** is an intuitive illustration of energy-level changes based on structural distortions in tin-based perovskites. Arrows indicate the shift in energy levels upon atom or small molecule substitution

the trends in these energies levels when the A cations are exchanged (Fig. 4c).

**Varying the X anion**. From Table 1 and Fig. 1 it is obvious that exchanging the halide leads to significant changes in both EA and IE. We discuss this finding using CsPbX$_3$ as example; the schematic energy diagram is shown in Fig. 4a. The energy of the CBM is mostly influenced by the position of the Pb,$p$ atomic level which is shifted upward when going from I to Br to Cl. Very likely this is a confinement effect, i.e., as the Pb-X distances decrease going from I to Br to Cl, an electron on a Pb atom is more confined and its energy increases.

The energy of the VBM is influenced by three competing effects. There is a significant downward shift of the X,$p$ level going from I to Br to Cl, which simply reflects increasing electronegativity. This is expected to cause a large downward shift of the VBM. This trend is lessened by two factors working in the opposite direction. Firstly, there is a shift upward of the Pb,$s$ level, which is the same confinement effect as discussed above. Secondly, the Pb,$s$/X,$p$ hybridization strength increases somewhat going from I to Br to Cl (see Supplementary Table 3). However, the downward shift of the X,$p$ level is still the dominating factor in determining the position of the VBM. A direct consequence of both the upward shift of the CBM and the downward shift of the VBM is the well-known substantial increase in the band gap going from I to Br to Cl.

**Varying the M cation**. Keeping the halide anion fixed and comparing Pb and Sn compounds in Table 1, it is evident that the IEs and EAs of Pb perovskites are larger than those of corresponding Sn ones. We analyze this trend using CsMI$_3$ (M = Pb, Sn) as example, shown in Fig. 4b. Replacing Pb with Sn, the atomic levels shift upwards, which is consistent with the smaller electronegativity of Sn. In absence of large changes in the anion levels, both the VBM and the CBM shift upward, i.e., both the IE and the EA of the Sn compounds are smaller than those of the corresponding Pb compounds. At the same time, we find that the splitting between s and p states in a Sn atom is smaller than in a Pb atom, which means that, going from Pb to Sn, the upward shift of the $s$ level is larger than that of the $p$ level; the consequence is that the VBM shifts upward more than the CBM. In other words, substituting Pb with Sn gives a strong reduction in the IE and a moderate reduction in the EA. The upward shift of the VBM is further enhanced by a slight increase in M,$s$/X,$p$ hybridization going from Pb to Sn (Supplementary Table 3).

This upward shift in VBM can be nicely observed in the experimental data, i.e. in Figs. 1 and 2, where clearly an additional feature appears at the band edge of the VB side for the Sn based

perovskites. Overall, the shifts of the VBM and CBM lead to a reduction of the band gaps of Sn compounds compared to their Pb analogs. There are three exceptions to this trend (FASnBr$_3$, MASnCl$_3$, and FASnCl$_3$) where we suggest that secondary effects, such as lattice distortions, play a larger role; these will be discussed next.

**Varying the A cation**. When changing the A cation, the IEs and EAs do not show a uniform trend. As mentioned earlier, the A site cation does not directly participate in the bonding and only influences the electronic structure indirectly via changing the volume of the AMX$_3$ lattice or by introducing distortion in the ideal perovskite structure.

An indicator for possible distortions of perovskite lattices is the commonly used Goldschmidt's tolerance factor[52] of $TF = \frac{r_A + r_X}{\sqrt{2}(r_M + r_X)}$, where $r_\square$ is the radius of the corresponding ion. It is commonly accepted that 3D perovskite structures form for $TF$ in the range of $0.8 < TF \leq 1$. In the lower part of this range the structures are distorted by tilting of the MX$_6$ octahedra, $TF = 1$ results a perfect cubic perovskite structure, and for $TF > 1$ or $TF < 0.8$ additional distortion of the octahedra can occur and alternative structures instead of 3D perovskites are possibly formed[53].

Structural deformations, including octahedral tilting and distortion of the octahedra in the MX$_3$ framework, reduce somewhat the hybridization between the M and X states throughout the crystal. This shifts the VBM and CBM downward, whereby the VBM and with it the IE is affected more because it is more sensitive to hybridization. Increasing the size of the A cation going from Cs to MA and FA also generally increases the volume (see Supplementary Notes 1–18 for structural information). This lowers the M levels somewhat (due to moderation of the confinement effect, see the discussion above). Again, this increases the IE and EA, but now the EA is affected most, as it is more sensitive to the M levels. In summary, both lattice distortion and volume expansion increase the IE and EA, where the former affects the IE most, and the latter the EA.

The interplay of these factors allows one to rationalize the variations in the IE and the EA of the lead-based perovskites. Here, varying the A cation within one halide class leads to relatively mild changes in volume and structures; hence, the IEs and EAs mostly vary only little. It is notable though that the EA of all FAPbX$_3$ compounds are larger by about 300 meV than their MA and Cs counterparts, which is due to the effect of increased volume described in the previous paragraph. Furthermore, MAPbI$_3$ shows an unusually low IE value which can be explained by its ideal $TF$ close to 1 (see Supplementary Table 2); it is therefore least affected by lattice distortion and will have the

highest degree of hybridization, resulting in an effective upshift of the VB; in contrast, $CsPbI_3$ and $FAPbI_3$ have larger IE due to a reduced hybridization. Indeed, these two compounds are well-known to be distorted and tend to form a 2D yellow phase at room temperature while the black phase is only accessible via a high temperature annealing step[15,54].

In contrast to the subtle changes in Pb perovskites, much larger variations in EAs and IEs are found in Sn perovskites. Likely, the larger influence of A-site substitution on Sn perovskites comes from the smaller ionic radius of Sn compared to Pb leading to the fact that Sn compounds have larger $TF$'s than their Pb counterparts; for Cs containing compounds the $TF$ becomes closer to 1, while Sn in combination with MA or FA leads to $TF$ values >1 (see Supplementary Table 2). Within the $ASnCl_3$ series in particular, the lattice distortion increases severely going from Cs to MA to FA, which means that the IE and EA increase, as depicted in Fig. 4c. As the IE is affected most, this means that the band gap increases significantly within this series via a downshift of the VB. Notably, also $FASnBr_3$ has a large $TF$ and is severely distorted. In fact, the large band gaps found for $FASnBr_3$, $MASnCl_3$, and $FASnCl_3$ go against the general trend we established previously, where we stated that the band gaps of Sn perovskites are generally smaller than those of Pb perovskites.

## General trends in the $AMX_3$ systems

Figure 5 shows a schematic diagram of all extracted energy level values sorted by their optical gaps (see Supplementary Fig. 1 for changes in energy levels sorted by IE's and EA's). The values are taken from Table 1, where the absolute errors of the respective measurements can be found. Overall, the IE (position of the VBM) varies more strongly than the EA and is determined by the hybridization between the M,$s$ and the X,$p$ states. We can identify two trends regarding changes in IE: it increases in energy going from I to Br to Cl due to the downshifting of the X,$p$ states while it also increases when switching from Sn to Pb, mainly due to the higher lying Sn,$s$ states compared to the Pb,$s$ states. The EA (energy of the CBM) is determined by the hybridization between the M,$p$ and the X,$s$ states, whereby the M,$p$ position plays the dominant role, as the X,$s$ level lies too low to affect the CBM much. The EA decreases when this M,$p$ level is shifted upwards, which happens when going from I to Br to Cl due to confinement effects, as well as when changing from Pb to Sn. Combining the trends for the IE and the EA, we find the band gap generally increases going from I to Br to Cl, and going from Sn to Pb. Compounds where the lattice is strongly distorted with respect to the ideal perovskite structure can however break any of the above mentioned trends.

In summary, by combining photoelectron spectroscopy measurements and DFT calculations, we provided a consistent and complete set of absolute energy levels of all primary tin- and lead-based halide perovskites. We clarified the physical origin of the trends observed in ionization energy, electron affinity, and band gaps by an elaborated analysis based on a simple tight-binding model. Important factors have been identified, which play key roles for determining the absolute energy levels. These are the effective atomic energy levels of metal cations and halide anions, their hybridization strength, as well as structural variations including size and distortion of the crystal lattice. This study therefore provides the basis to optimize interfaces in optoelectronic applications and to further engineer energy levels of more complex (mixed) halide perovskites.

## Methods

**Computational procedures**. *DFT calculations*: all calculations on metal halide perovskites are performed using density-functional theory (DFT) and the projector-augmented wave (PAW)/plane wave techniques, as implemented in the Vienna ab-initio simulation package (VASP)[55,56]. The Perdew, Burke, and Ernzerhof (PBE)[57] functional is used for geometry optimizations. An energy cutoff of 500 eV and a $k$-point scheme of $6 \times 6 \times 4$, $6 \times 4 \times 6$, $6 \times 6 \times 6$ are used for tetragonal, orthorhombic, and cubic structures, respectively, and energy and force convergence parameters are set to 0.1 meV and 20 meV/Å, respectively.

The hybrid functional PBE0[58] is used to calculate the density of states (DOS), with a small smearing parameter of 0.02 eV in a Gaussian smearing scheme. This small smearing parameter allows accurate identification of the onsets of valence and conduction bands in the DOS, as needed for the evaluation of the PES experiments, described in Fig. 2. Here, to reduce the computational cost, we use a reduced $k$-point scheme, namely, $6 \times 6 \times 4$, $6 \times 4 \times 6$, and $3 \times 3 \times 3$ for tetragonal, orthorhombic, and cubic structures. We have adopted tetragonal cells for all MA and FA perovskites and orthorhombic cells for the Cs perovskites. The underlying reasons are explained in the discussion of the Supplementary Figs. 3 and 4. We included spin-orbit-coupling (SOC) in our DFT calculations for the DOSs of all 18 perovskites. These DOSs were used for the fitting described in Fig. 2 in the main text. To start the tight binding analysis, a separate set of DFT calculations of Cs perovskites with cubic symmetry were performed without SOC, enabling a more straightforward identification of the relevant levels on the basis of symmetry (see Methods: Tight binding analysis). SOC is then added subsequently to the tight-binding model.

*COHP analysis*: to analyze the electronic structure and bonding in halide perovskites, we calculate the density of states (DOS), the partial density of states (PDOS) and the crystal orbital Hamiltonian population (COHP) with the Lobster 2.2.1 code[59–61]. This involves a transformation of the plane wave basis set used by VASP, to a localized basis set of Slater-type orbitals (STO).

The PDOS is defined as:

$$\text{PDOS}_i(E) = \sum_n |c_i^n|^2 \delta(E - E_n), \tag{1}$$

where $c_i^n$ are the coefficients associated with the atomic orbitals $\phi_i$ in a molecular orbital $\psi_n = \sum_i c_i^n \phi_i$. The COHP is defined as:

$$-\text{COHP}_{ij}(E) = H_{ij} \sum_n c_i^n c_j^{*n} \delta(E - E_n), \tag{2}$$

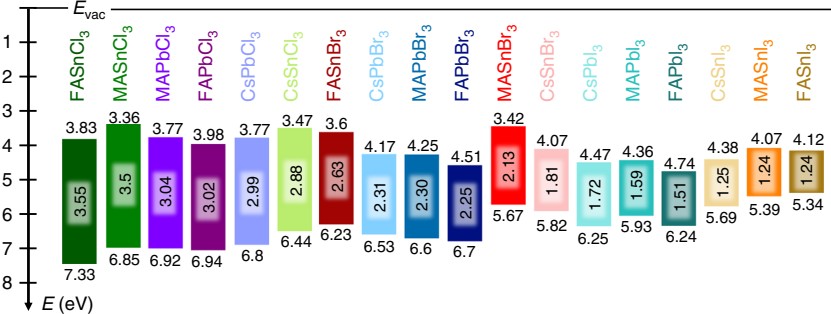

**Fig. 5** Schematic energy level diagram of the 18 metal halide perovskites. The respective IE and EA values as well as the optical gaps (all in eV) are taken from Table 1; compounds are sorted in order of decreasing band gap. Minor deviations between the optical gap and the difference between IE and EA stem from the fact that one is extracted from the UV-vis measurements and the other from the PES investigations, which are furthermore averaged over 3 samples each; the corresponding error bars are given in Table 1

where $H_{ij}$ is the Hamiltonian matrix element between the atomic orbitals $\phi_i$ and $\phi_j$. For positive values of $-COHP_{ij}(E)$ the electronic interaction between the atomic orbitals $i$ and $j$ is bonding, negative values of $-COHP_{ij}(E)$ symbolize an anti-bonding interaction, while a zero value designates a non-bonding interaction. COHP has proved to be very powerful in the analysis of magnetism[62], phase stability[63], and catalytic reactivities[64,65] of solid state materials.

*Tight binding analysis*: we use a nearest neighbor tight-binding model for the $Pm\text{-}3m$ cubic structure of perovskites, as described by Boyer-Richard et al.[51]. The model has six parameters: the on-site energies of the $s$ and $p$ levels of the X and M atoms, $E_{X,s}$, $E_{X,p}$, $E_{M,s}$, and $E_{M,p}$, and the two hopping parameters (hybridization strengths) $V_{M,p\text{-}X,s}$ and $V_{M,s\text{-}X,p}$. The VBM and CBM are situated at the R-point of the Brillouin zone, so we restrict the bonding analysis to states at the R-point. The energy levels $E_{1-4}$, see Fig. 3c and Supplementary Fig. 2, are then given by the expressions

$$E_{1,4} = \frac{E_{X,s} + E_{M,p}}{2} \pm \frac{\left[(E_{M,p} - E_{X,s})^2 + 16V_{M,p\text{-}X,s}^2\right]^{\frac{1}{2}}}{2} \quad (3)$$

$$E_{2,3} = \frac{E_{M,s} + E_{X,p}}{2} \pm \frac{\left[(E_{M,s} - E_{X,p})^2 + 48V_{M,p\text{-}X,s}^2\right]^{\frac{1}{2}}}{2} \quad (4)$$

Here, the + signs give the CBM $E_1$ and the VBM $E_2$, while the – signs correspond to levels deep in the valence band $E_3$ and $E_4$. The energies $E_{1,4}$ and $E_{2,3}$ can be identified in the DFT spectrum at the R-point. They correspond to states with $R_4^+(3)$ and $R_1^+(1)$ symmetry, respectively.

The two anion levels, $E_{X,s}$ and $E_{X,p}$, can be identified in the DFT spectrum at $\Gamma$-point. They correspond to non-bonding halide states with $\Gamma_3^+(2)$ and $\Gamma_5^-(3)$ symmetry, respectively. Alternatively, we can obtain $E_{X,p}$ from the DFT spectrum at the R-point. In a nearest neighbor tight-binding model, one should find an eight-fold degenerate level of non-bonding halide $p$ states. With non-nearest neighbor interactions this degeneracy splits up into a doublet and two triplets with $R_3^+(2)$, $R_4^+(3)$, and $R_5^+(3)$ symmetry, respectively. Indeed, in the DFT spectrum we find a splitting of about 1 eV between the $R_4^+(3)$, and the $R_5^+(3)$ triplet states. If we assume that the next-nearest neighbor interaction between anion $p$-states is responsible for this splitting, then the $R_4^+(3)$ and $R_5^+(3)$ levels are split by a single hopping-matrix element[66], and the average of the two corresponds to the anion $p$ level $E_{X,p}$. Indeed, we find a difference smaller than 0.1 eV between $E_{X,p}$ extracted from the DFT spectra of the R-point and the $\Gamma$-point.

The remaining four parameters, $E_{M,s}$, $E_{M,p}$, $V_{M,p\text{-}X,s}$, $V_{M,s\text{-}X,p}$, can then be extracted from Eqs. (3) and (4). For simplicity, the symmetry analysis described above is performed in absence of spin-orbit coupling (SOC). SOC is largest for the cation $p$ states, where it is easily included by subtracting $\frac{2}{3}\Delta_{SOC}$ from $E_{M,p}$ in Eqs. (3) and (4), with $\Delta_{SOC}$ the SOC splitting[23,51]. More details can be found in the Supplementary Methods. It should be noted that the experimental values of IE and EA from Table 1 are used to correct the DFT energy values of $E_2$ (VBM) and $E_1$ (CBM); all other levels extracted from DFT are shifted accordingly. All relevant energy levels and hopping parameters are summarized in the Supplementary Table 3.

**Experimental procedures**. *Sample preparation*: investigating perovskites poses several challenges. It is often reported that variations in processing can lead to sample-to-sample variation, either due to variations in film stoichiometry or partial or complete transition into different crystal structures. For some examples and a more detailed discussion of these effects we refer to the Supplementary Figs. 7–10. Such variations can lead to changes in work function (Wf)[67,68] ionization energy[31–33], or the band gap[54,69]. Therefore, in this study great care was taken to ensure the preparation of representative perovskite films. Typically, for each composition dozens of samples were tested using a variety of preparation methods. Most films were prepared by solution processing and variations include the choice of solvent, co-solution vs. sequential deposition, spin speed, antisolvent treatment, and annealing procedure. In some cases thermal co-evaporation was used; this was especially necessary for various Cs containing compounds, since solubility of CsCl, and to some extend CsBr, provided major challenges. Films were tested and optimized with respect to their absorption properties, film morphology (via Scanning Electron Microscopy, SEM), crystal structure (via X-Ray Diffraction, XRD) and films composition (via X-ray Photoelectron Spectroscopy, XPS). With XPS, the films were also checked for unwanted oxidation states, e.g. the presence of signal originating from $Sn^{4+}$ or $Pb^0$. The corresponding absorption, SEM, XRD, and XPS measurements for representative samples can be found in the Supplementary Notes 1–18.

*Samples were prepared* on top of PEDOT: PSS (Clevios P VP Al 4083, Heraeus) covered indium tin oxide substrates (ITO from Thin Film Devices). The 40 nm thick PEDOT:PSS layer was employed in order to passivate the ITO surface which is otherwise known to undergo detrimental reactions with the perovskite at the interface[70]. The perovskite solution processing was performed under nitrogen atmosphere, always using a 1:1 molar ratio of the precursor salts in either DMF or DMSO. A list of materials, purity and vendors can be found in the Supplementary Table 5. In some cases, an orthogonal solvent was used during the spin coating procedure to induce faster crystallization. Thermal evaporation was used for a set of Cs containing samples (CsMCl₃, and CsMBr₃). Co-evaporation was done using a

molar ratio of the precursors close to 1:1. Some of the samples were annealed in vacuum. A detailed description of the individual preparation procedures, such as concentration, spin speed, and annealing time, is listed in the Supplementary Table 6.

Photoelectron spectroscopy measurements were performed in a custom built multi-chamber ultra-high vacuum setup. Thermally evaporated samples were transferred directly into the measurement chamber without breaking the vacuum, while solution processed films were transferred though nitrogen atmosphere; no sample was air exposed at any time and samples were measured within 24 h after preparation. For the measurement of the occupied DOS and work function via UV photoelectron spectroscopy, a monochromatic He plasma source (VUV 5 k, Scienta Omicron) at an excitation energy of 21.22 eV was used in combination with a hemispherical electron analyser (Phoibos 100, Specs) at an electron pass energy of 2 eV; a sample bias of $-8$ V was applied during measurements to observe the high energy cutoff. The experimental resolution at this low pass energy setting is only determined by thermal broadening and is in the range of 100 meV ($\Delta E = 4k_BT$). For some of the samples, additional Kelvin Probe (KP) measurements (KP6500, McAllister) in vacuum were performed to compare the Wf measured under illumination (UPS) to the one in the dark (KP) since changes in surface dipole have been reported in literature[71]. No significant difference was found between the measurements, except in some samples of CsPbCl₃ and CsSnCl₃. Here a light dependence change in Wf was observed, with the Wf being lowered by about 0.5 eV due to illumination. After a series of tests with different sample treatments it was found that a moderate annealing in vacuum to 60–80 °C made the effect vanish (resulting difference between KP and UPS $\le$ 60 meV).

Measurements of the unoccupied DOS were performed by inverse photoelectron spectroscopy. Here, a low energy electron gun (ELG-2, Kimball) was used at 2 µA emission current together with a bandpass photon detector (SrF₂/ NaCl bandpass, IPES 2000, Omnivac). The energy resolution, as determined from the width of an Ag Fermi edge, is approximately 600 meV. Since the electron bombardment during IPES measurements can be harmful to a sample surface, it was always performed after the UPS and XPS measurements were finished. The samples were re-checked via UPS afterwards to ensure no severe change in the DOS was induced by the IPES measurements. Typically, sample containing Cs and FA as cations were very stable, however some of the MA samples (MASnI₃, MASnBr₃, MASnCl₃, and MAPbI₃) could not be measured for more than 5–10 minutes by IPES before a change in the DOS occurred.

All other experimental characterization methods, such as SEM, UV-vis, XPS, and XRD, are described in the Supplementary Methods section.

## Data availability

All energy level values of the individual samples, which were used for obtaining the averaged values throughout this paper, are included in the Supplementary Table 6. All remaining data sets supporting the findings of the work, including DFT as well as experimental results, are available from the authors upon reasonable request.

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

## Acknowledgements

This work was in parts supported by the ministry of Science of the state of NRW within the PeroBOOST (EFRE) project. S.O. acknowledges funding by the Eleonore Trefftz Programme for Visiting Women Professors at TU Dresden as well as the DFG Project MUJUPO. S.T. and J.J. acknowledge funding by the Computational Sciences for Energy Research (CSER) tenure track program of Shell and NWO (Project Number 15CST04-2), the Netherlands. S.T. and I.T. thank the NWO for access to the Dutch national high-performance computing facilities (Cartesius).

## Author contributions

The project was conceived, planned, and coordinated by S.T. and S.O.; Samples were prepared and characterized by I.S. and S.O., K.M. supervised part of the project; S.T. performed DFT calculations; G.B. performed the tight binding analysis; J.J. and I.T. performed the COHP analysis; S.T., G.B. and S.O. provided the interpretation of the results. S.O. and S.T. wrote the first version of the manuscript, and all authors contributed to the final version.

## Additional information

**Competing interests:** The authors declare no competing interests.

