## [Peer Review File · Nature Communications]

Reviewers' comments:

Reviewer #1 (Remarks to the Author):

In this work, using UPS/IPES Tao et al. provide the experimentally determined energy levels of various compositions of Sn and Pb based perovskites with varying cations (MA, FA, Cs) and anions (I, Br, Cl). By combining these results with theoretical calculations, the authors explain the physical origin of the trends observed in IE, EA, and band gaps. I expect this work to have a broad impact on the community working with perovskite related materials and device applications. However, a few points (see below) should be thoroughly addressed before it can be accepted.

1. As the authors mentioned in their introduction, (1) the origins underlying the perovskite material bandgap and (2) the positions of VBM and CBM of these materials as a function of compositions are not only fundamentally highly interesting, but are also needed to develop strategies for tailoring desired optoelectronic properties and to optimally match perovskite energy levels to contacts and extraction layers for efficient charge transport through a device. The following references are highly relevant to this topic and should be discussed to provide readers with a comprehensive picture of the recent development in the field towards this direction:

[1] Electronic structure of the CsPbBr₃/polytriarylamine (PTAA) system. *J. Appl. Phys.* 121 (2017) 035304.

[2] Fermi level, work function and vacuum level. *Mater. Horiz.* 3 (2016) 7-10.

[3] Energy Level Alignment at Interfaces in Metal Halide Perovskite Solar Cells. *Adv. Mater. Interfaces* 5 (2018) 1800260.

[4] Recent Advances in Energetics of Metal Halide Perovskite Interfaces. *Adv. Mater. Interfaces* 4 (2017) 1600694.

[5] Energy Level Offsets at Lead Halide Perovskite/Organic Hybrid Interfaces and Their Impacts on Charge Separation. *Adv. Mater. Interfaces* 2 (2015) 1400528.

2. The presented section 9 in the supporting information is helpful. However, I notice that the CsSnI₃ perovskite film morphology shows large pinholes and cracks (page 17, supporting information). The UPS spectrum for this perovskite would correspond to a convoluted signal of perovskite + ITO. The XPS spectra shown for MASnI₃, FASnI₃, and several other perovskites show solvent and substrate related peaks in the C 1s spectra. It is important for the authors to comment on how much influence this would have on the conclusions that the authors extracted from the UPS analysis.

3. How many samples were measured for the spectra shown in Figure 1 and the extracted energies shown in Table 1? As the authors mentioned, slight variations in perovskite stoichiometry can result in variations in energy levels. How much is the standard deviation due to sample variation compared to instrumental errors of UPS/IPES? Did the authors perform any control experiments to quantify these effects (i.e., variations in perovskite stoichiometry)?

4. On the basis of Ref. 30 [J. Phys. Chem. Lett. 7 (2016) 2722], I would suggest the authors to display Figure 1 also in semi-log plots.

5. I would suggest that the authors provide the Fermi level positions in Figures 1, 5, and S1 extracted from the secondary electron cutoff onsets in UPS.

6. On page 5, it is mentioned that the “DOS broadened spectra are fitted with Gaussian peaks and these peaks do not have a direct physical meaning; they do not represent features in the perovskites DOS, but are chosen such that they are able to consistently describe all perovskite compositions.” This methodology can potentially be problematic because mathematically it is possible to fit any spectra with any arbitrary number of superposed Gaussian peaks. Would it be possible here to elaborate on a more physically-meaningful methodology such as considering the individual orbital contributions of different elements in perovskites? For example, similar to what is shown in Figure 3a, would it be possible to show the partial density of states in Figure 2 as well? This would give better direct assignments of different contributions to the different peaks being fitted in the “Calculated DOS broadened” and “Measured DOS”.

7. In the subsequent paragraph, the authors mentioned that “Our procedure proves to be very robust for the VB region and therefore the position of the VBM can be accurately determined.” The report in Ref. 30. J. Phys. Chem. Lett. 7 (2016) 2722 describes that near the VBM region in lead-based perovskites, the DOS contribution comes from the weak Pb 6s, Pb 6p, and halide-related orbital contributions. A strategy is to plot the UPS intensity in semi-logarithmic scales which can help determine the VBM more accurately. Would it be possible for the authors to provide a similar Table 1, but with the values extracted from the semi-log scale analysis? It will be important for readers to know the differences in the IE and EA values extracted depending on these two proposed methodologies.

8. On the basis of absorbance spectra, some perovskites show strong excitonic features (e.g., CsPbBr₃, MAPbBr₃, FAPbBr₃, CsPbCl₃, MAPbCl₃, and FAPbCl₃). The authors extracted the optical bandgaps and on the basis of IE and EA the authors would be able to quantify the transport bandgaps. This work may provide further insight if the authors can add a section analyzing the excitonic properties in these materials. It is interesting to see that Sn based perovskites do not show

the characteristic excitonic features in absorbance. Please comment on that and add relevant references.

Reviewer #2 (Remarks to the Author):

The manuscript "Absolute energy level positions in tin and lead based halide perovskites" is a joint experimental/computational investigation of to establish a band alignment comparison amongst all halide perovskites for energy extraction in optoelectronic devices. The authors determine the ionization energy and electron affinity values for all primary tin and lead based halide perovskites using photoelectron spectroscopy and first-principles calculations. They use tight-binding analysis to characterize the key energy level variations. The authors find that the energy level variations are primarily determined by the relative positions of the atomic energy levels of metal cations and halide anions. Cation-anion interaction strength plays a secondary role. These results are an important step towards rational design rules for perovskite devices and is of interest to a broad community. This study is a systematic and through study that is worthy of publication in Nature communications after the following are addressed:

- 1.) The authors should make clearer when they use spin orbit coupling (SOC) in their DFT calculations. Sometimes it is implied that it is used, sometimes it is not mentioned.
- 2.) vdW's interactions appear not to have been taken into account in geometry optimization. While having very little direct effect on the electronic structure. They have been shown to be critical to getting the correct geometry (eg. lattice parameters and octohedral tilting) which does have an influence on the electronic structure. Can the authors please address why it was not included and maybe take one test material (with an organic component) and test that it makes little difference?
- 3.) There is a typo in the numerator of the goldschmidt tolerance factor in both the main manuscript and SI: r_m should be r_x . It looks like the values were calculated correctly though.
- 4.) While there is no extensive study of the band alignment of all of the Sn and Pb halide perovskite materials there have been several of MAPbI₃ (most of which the authors cite), but it could be helpful if the authors compare their values for the systems that have been done and put them in context. For instance: <http://dx.doi.org/10.1016/j.solener.2015.07.040>
- 5.) Perhaps also a discussion how accurate the absolute values of the IE and EA are compared to the relative values are.

Response Letter

We thank the reviewers for their careful consideration of our manuscript “Absolute energy level positions in tin and lead based halide perovskites”. We are grateful for the very positive evaluation and insightful suggestions. In considering and addressing the reviewer’s comments, the revised manuscript is now conveying several of the raised issues more clearly. As requested by Nature Communication, we also added a Data Availability Statement. Please find below our detailed response to the reviewers remarks in blue. Changes made to the manuscript are highlighted here as well as in the revised manuscript and SI.

Comments to Reviewer #1:

We thank the reviewer for his/her acknowledgement of the importance and urgency of our work and for his/her deep insights and useful suggestions regarding the experimental aspects of this material class.

1. As the authors mentioned in their introduction, (1) the origins underlying the perovskite material bandgap and (2) the positions of VBM and CBM of these materials as a function of compositions are not only fundamentally highly interesting, but are also needed to develop strategies for tailoring desired optoelectronic properties and to optimally match perovskite energy levels to contacts and extraction layers for efficient charge transport through a device. The following references are highly relevant to this topic and should be discussed to provide readers with a comprehensive picture of the recent development in the field towards this direction: [1] Electronic structure of the CsPbBr₃/polytriarylamine (PTAA) system. *J. Appl. Phys.* 121 (2017) 035304. [2] Fermi level, work function and vacuum level. *Mater. Horiz.* 3 (2016) 7-10. [3] Energy Level Alignment at Interfaces in Metal Halide Perovskite Solar Cells. *Adv. Mater. Interfaces* 5 (2018) 1800260. [4] Recent Advances in Energetics of Metal Halide Perovskite Interfaces. *Adv. Mater. Interfaces* 4 (2017) 1600694. [5] Energy Level Offsets at Lead Halide Perovskite/Organic Hybrid Interfaces and Their Impacts on Charge Separation. *Adv. Mater. Interfaces* 2 (2015) 1400528.

→ We agree with the reviewer that the alignment of perovskite energy levels to those in transport and contact layers is highly important for designing efficient devices and we have included the references mentioned by the reviewer to the introduction. As these device issues are not topics further addressed in this work, we have however not included an extensive discussion of the respective developments in this field.
2. The presented section 9 in the supporting information is helpful. However, I notice that the CsSnI₃ perovskite film morphology shows large pinholes and cracks (page 17, supporting information). The UPS spectrum for this perovskite would correspond to a convoluted signal of perovskite + ITO. The XPS spectra shown for MASnI₃, FASnI₃, and several other perovskites show solvent and substrate related peaks in the C 1s spectra. It is important for the authors to comment on how much influence this would have on the conclusions that the authors extracted from the UPS analysis.

→ We thank the reviewer for mentioning the helpfulness of this section 9 in the Supplementary Information, showing the extensive material analysis. Indeed, in a few cases uncovered surface or traces of solvents are present and the resulting carbon signals are notable in the XPS spectrum as indicated in section 9. In principle, this might lead to additional DOS features coming from the substrate (which is PEDOT:PSS in these measurements). However, its density of states is rather low at small binding energies, as shown in the graph below. Here a comparison of CsSnI₃ and a pristine PEDOT:PSS sample is shown. In a partially covered sample, this respective PEDOT:PSS

signal would be further reduced to a fraction of this intensity and not significantly influence the valence band and conduction band DOS. However, deeper in the bands (i.e. beyond ± 4 eV) PEDOT:PSS as well as solvent-signals could add more significantly to the DOS. This could be one of the reasons why in literature sometimes the typical shape of the DOS of a perovskite as seen below is masked by a gradual slope, see e.g. DOI:10.1116/1.4915499. In our work, we find good agreement between measurement and DFT over a wide range.

Further, it is important to note that additional signals at the band edges would result in erroneous results only if the energy levels were extracted directly from the experiment (by a linear or log scale extrapolation). However, we read out the onset values from the DFT calculated density of states (DOS), by aligning the material's distinct DOS features from theory (DFT) and from experiment (UPS/IPES). This way, any additional features have no influence on the results.

To clarify this point, we have adjusted the phrasing on page 3 of the manuscript:

For example, the imperfect surface or defects in the film can introduce additional states at the band edges while an insufficient coverage can lead to substrate features appearing in the measured DOS. Furthermore, the typically low DOS of perovskites at the onsets can be missed with common fitting procedures. Such issues can be circumvented to a certain degree by fitting the experimental DOSs...

- 3a How many samples were measured for the spectra shown in Figure 1 and the extracted energies shown in Table 1?
- Figure 1 presents specific measurements of individual samples, which we consider to be representative for the materials. Table 1 represents values averaged over three samples; those measurements are shown in the supplementary material, Figure S6. In the revised manuscript, we have made this clearer:
- On page 2: Figure 1 shows UPS and IPES measurements on representative films of all 18 tin- and lead-based AMX_3 perovskites
- Caption Fig. 1: Representative UPS and IPES measurements of all 18 metal halide perovskite systems
- Table 1: All values, which are given in eV, are averaged over 3 samples each; the error bars represent the spread over these samples
- 3b As the authors mentioned, slight variations in perovskite stoichiometry can result in variations in energy levels. Did the authors perform any control experiments to quantify these effects (i.e., variations in perovskite stoichiometry)?
- We share the reviewer's opinion that variations in film composition can lead to variations in electronic structure. The three samples chosen for our analysis (see point 3a) are selected after

extensive material optimization processes for each of the 18 perovskite systems as detailed in the methods section.

To elaborate on this a bit more: We started off by optimizing preparation conditions with respect to film morphology (SEM measurements) and absorption behavior (UV-vis) during which 10-20 preparation conditions were tested. For solution processing we kept equal molar ratios of the precursors in all cases, even though due to solubility issues this is not necessarily represented in the final film. For co-evaporation also the respective evaporation rates were slightly modified (see Table S6). The most promising films were further characterized by XPS, XRD, and PES (8-10 additional samples) to identify film composition, phase purity, appearance of gap states, etc. which formed the basis for selecting the final films shown for the investigations here.

Through this series of experiments, we are able to quantify in all cases to various degrees how changes in composition effect the electronic structure of the specific perovskites. The result are very similar to what we have previously published for MAPbI₃ (DOI: 10.1002/adma.201503406), meaning that an increase in the metal precursor leads to an increase in IE and an increase of the MA/FA/Cs precursor decreases the IE. The strength of this effect is quite different for the different systems, as is shown for a few examples in the graph below.

Please note that the values shown here correspond to a simple linear readout, since those test samples were not evaluated using the approach via the fit to the DFT data. This is why absolute values, even for the correct stoichiometry, differ from the ones in the paper.

These results are merely based on the preliminary testing, and no systematic variations were conducted. The results presented throughout this paper are on samples with appropriate stoichiometry. Therefore, the discussion presented here is not included in the manuscript and only provides an answer to the reviewers question regarding the analysis of the quantitative effects of stoichiometry variation.

3c How much is the standard deviation due to sample variation compared to instrumental errors of UPS/IPES?

- We have evaluated three representative samples each and from such a limited data set we cannot extract a standard deviation. In Table 1 we show error bars that give an impression of the sample-to-sample variation of our samples, which are typically in the range of ± 0.1 eV. The instrumental errors of UPS and IPES are mentioned in the methods section and do not play a significant role in this work, since we do not extract the band onsets directly from the measured data. The experimental broadening merely plays a role for the choice of Gaussian broadening we apply to the DFT (see Figure 2), which is however again not relevant for the onset extraction which is done from the non-broadened DFT spectra.
4. On the basis of Ref. 30 [J. Phys. Chem. Lett. 7 (2016) 2722], I would suggest the authors to display Figure 1 also in semi-log plots.
- For our reply, we refer to remark 7, which deals with the same issue.
5. I would suggest that the authors provide the Fermi level positions in Figures 1, 5, and S1 extracted from the secondary electron cutoff onsets in UPS.
- As suggested by the reviewer, we have added the Fermi level positions to Figure 1 and marked them by triangles. However, in Figures 5 and S1 we are not able to do this, since the energy levels plotted there (ionization energy (IE) and electron affinity (EA)) are averages of the values extracted from three samples. While the IE and EA are intrinsic material properties, the Fermi level position is not, as it depends upon the doping level. Relatively small changes in the latter will move the Fermi level through the band gap, so preparation conditions affect it. For the interested reader, all work functions are provided in Table S6, and also the UPS/IPES spectra plotted in the Figure S6 are plotted with respect to the Fermi level.
6. On page 5, it is mentioned that the “DOS broadened spectra are fitted with Gaussian peaks and these peaks do not have a direct physical meaning; they do not represent features in the perovskites DOS, but are chosen such that they are able to consistently describe all perovskite compositions.” This methodology can potentially be problematic because mathematically it is possible to fit any spectra with any arbitrary number of superposed Gaussian peaks. Would it be possible here to elaborate on a more physically-meaningful methodology such as considering the individual orbital contributions of different elements in perovskites? For example, similar to what is shown in Figure 3a, would it be possible to show the partial density of states in Figure 2 as well? This would give better direct assignments of different contributions to the different peaks being fitted in the “Calculated DOS broadened” and “Measured DOS”.
- We thank the reviewer for pointing out that this section was not sufficiently clear and could benefit from presenting additional information. Indeed, enough Gaussian peaks are able to fit any shape. It should be noted though that we only exploit the fitting to align the DFT spectrum with the measured spectrum; therefore, as long as a single set of peaks is found that mathematically describes both these spectra well, the alignment is robust. However, saying here that there is no direct physical meaning could be misleading and we thank the reviewer for this useful advice. We have added a subfigure in Figure 2 showing the partial DOS of these two material systems. Most of the Gaussian peaks used for fitting can be associated with features from the partial DOS. Specifically, this is the case if the total DOS in a particular energy region is dominated by one partial DOS. If multiple partial DOSs overlap strongly, the correlation with the Gaussian peaks is less direct.
- In addition to modifying Fig. 2, we have removed the claim of *these peaks do not have a direct physical meaning* and modified the description on page 5 which now reads:
To be able to match theory and experiment, these broadened DFT spectra are then fitted to a linear combination of Gaussian peaks; these are chosen such that they allow for a robust fit and

are able to consistently describe the various perovskite compositions. The peaks roughly correspond to the partial DOS features, as can be seen when comparing them to the features in the third panel.

7. In the subsequent paragraph, the authors mentioned that “Our procedure proves to be very robust for the VB region and therefore the position of the VBM can be accurately determined.” The report in Ref. 30. J. Phys. Chem. Lett. 7 (2016) 2722 describes that near the VBM region in lead-based perovskites, the DOS contribution comes from the weak Pb 6s, Pb 6p, and halide-related orbital contributions. A strategy is to plot the UPS intensity in semi-logarithmic scales which can help determine the VBM more accurately. Would it be possible for the authors to provide a similar Table 1, but with the values extracted from the semi-log scale analysis? It will be important for readers to know the differences in the IE and EA values extracted depending on these two proposed methodologies.

→ This issue raised by the reviewer is certainly an important point. The authors discussed this point extensively during the research and we decided not to show the IE/EA data sets extracted by either linear or log scale extrapolation. Our reasons for this decision are the following. Discussing the advantages and disadvantages of different techniques for extracting IE/EAs from PES data would be rather technical and take the focus away from the main messages of the paper to the broad readership, which is to identify the energy level positions of a wide range of lead and tin halide perovskites, and analyze the trends in these positions. In this paper we introduce an elegant method based on the alignment of experimental data with DFT spectra. Most non-specialist readers would likely be confused when presented with three different data sets from three different methods.

We do however agree that this issue is of interest to researchers working in the specific area of photoelectron spectroscopy and perovskites. We have prepared a comparison for the reviewer in this reply, plotting the values of IE and EA, using for extrapolation a linear scale, a log scale, and the approach introduced in this paper. The data sets and extracted values are shown in the additional document provided for the reviewers (called *Lin Log Comparison*), while the graph below displays the relation between the optical gap and the UPS-IPES extracted gap, employing these three different methods; the grey solid diagonal line indicates the expected trend. Both the linear and the log scale methods underestimate the band gap for most perovskite compounds, where the underestimation by the log scale method is much more pronounced.

We believe that this is due to a combination of three effects. (i) Experimental broadening, which is ~ 100 meV for UPS and ~ 600 meV for IPES, smears out the DOS into the band gap and therefore narrows the band gap. (ii) While lead-iodide compounds have a rather steep onset of the VB, in compounds containing Sn/Br/Cl it rises much more gently. Therefore, a log scale representation will likely lead to a wider tail of states in the band gap. (iii) Defect states can be present in the bandgap or at the band edges and obscure the band onsets. While some of the perovskites are known to be very defect tolerant, others are not, and in those cases we observed strong variations in log scale band onsets, as shown below. Here, measurements of the defect tolerant MAPbI_3 as well as of the defect intolerant MASnCl_3 are presented.

Without knowing the intrinsic shape of the bulk DOS of the material from DFT, it is not possible to determine whether states at the band edges belong to the perovskite or originate from defect states. Not only bulk defects introduce states but, as already mentioned in the paper, PES is a highly surface sensitive technique. The presence of a surface modifies the DOS and introduces additional states depending on the surface termination; therefore, it is likely that additional states

are present in many of the measurements. This is why we consider the method introduced in this paper to be very helpful, as it does not take into account these additional states.

The presence of defects states at band edges can also be shown in DFT calculations. Below we show as an example MAPbBr₃ (plotted in a semi-log scale). Here, the blue curve represents the calculated and broadened VB DOS of a perfect crystal while for the red curve an MA deficiency was introduced (one out of four MA molecules is missing), leading to the presence of gap states.

Regarding the changes made to the manuscript: So far, the lin/log discussion is not included in the paper and we only show the results here to convey the complexity of the problem to the reviewers; as mentioned above, we believe this discussion would take the focus too far away from the topic of the paper. If, however, the reviewer is convinced it should be included in this submission, we are willing to add a section to the Supplementary Information.

8. On the basis of absorbance spectra, some perovskites show strong excitonic features (e.g., CsPbBr₃, MAPbBr₃, FAPbBr₃, CsPbCl₃, MAPbCl₃, and FAPbCl₃). The authors extracted the optical bandgaps and on the basis of IE and EA the authors would be able to quantify the transport bandgaps. This work may provide further insight if the authors can add a section analyzing the excitonic properties in these materials. It is interesting to see that Sn based perovskites do not show the characteristic excitonic features in absorbance. Please comment on that and add relevant references.

→ We thank the reviewer for his/her sharp observations, in particular of the difference between the excitonic features of the Sn and the Pb perovskites. We acknowledge that a detailed analysis of the optical properties of these 18 perovskites is of interest and could result in further important insights.

Note however, that the exciton binding energies for 3D Pb perovskites, extracted previously from experiments, Refs. 43-45, are small, i.e., $\lesssim 0.05$ eV. In Ref. 46 it was already observed that the exciton binding energy in a Sn perovskite seems to be an order of magnitude smaller than that in the corresponding Pb compound. In any case, such small energies do not play an important role in determining the band onsets, which is what the present paper is about. The excitonic problem deserves a systematic study by itself, which we feel is outside the scope of the present paper.

Reviewer #2

We thank the reviewer for his/her appreciation of our work and the recommendation of a publication in Nature Communications. We highly appreciate his/her careful evaluation and suggestions for improvement of the paper, especially with respect to the theory aspects.

1. The authors should make clearer when they use spin orbit coupling (SOC) in their DFT calculations. Sometimes it is implied that it is used, sometimes it is not mentioned.

→ We included spin-orbit-coupling (SOC) in our DFT calculations for the DOSs of all 18 perovskites. These DOSs were used for the fitting described in Figure 2 in the main text. For the tight binding analysis, a separate set of DFT calculations of Cs perovskites with cubic symmetry were performed without SOC, enabling a more straightforward identification of the relevant levels on the basis of symmetry (see Methods section, Tight binding analysis). SOC is then added subsequently to the tight-binding model.

We have now made this point clearer in the revised manuscript by adding the following text in the Methods section:

We included spin-orbit-coupling (SOC) in our DFT calculations for the DOSs of all 18 perovskites. These DOSs were used for the fitting described in Figure 2 in the main text. For the tight binding analysis, a separate set of DFT calculations of Cs perovskites with cubic symmetry were performed without SOC, enabling a more straightforward identification of the relevant levels on the basis of symmetry (see Methods: Tight binding analysis). SOC is then added subsequently to the tight-binding model.

2. vdW's interactions appear not to have been taken into account in geometry optimization. While having very little direct effect on the electronic structure. They have been shown to be critical to getting the correct geometry (eg. lattice parameters and octohedral tilting) which does have an influence on the electronic structure. Can the authors please address why it was not included and maybe take one test material (with an organic component) and test that it makes little difference?

→ We indeed did not include vdW corrections in our calculations. As correctly stated by the reviewer, the inclusion of such corrections does not directly affect the calculated DOS.

In a previous publication, we tested vdW corrections produced by DFT-D3 (Grimme method) and DFT-TS (Tkatchenko-Scheffler method). We found that neither of these two methods gave a uniform improvement in the structure of the perovskites under investigation there. For instance, the lattice parameters of some of the compounds were overestimated, while those of others were underestimated. Details can be found in the method section of:

<https://www.nature.com/articles/s41598-017-14435-4> . On the basis of our previous experiences, we therefore did not include vdW corrections in our calculations here.

The PBE functional tends to slightly overestimate the lattice parameters (1-2%) of perovskites, compared to experiment (as it does in many other materials). This mainly influences the band gap, which increases by 30 - 80 meV, the DOS is little affected by this small change in lattice parameters. Most importantly, our fitting procedure only makes use of the shape and relative positions of the main peaks of the DOSs (Figure 2). The variations introduced by changing the lattice parameter by a few percent, are smaller than the experimental sample-to-sample variations (in the range of ± 100 meV in main text Table 1). Therefore, we can disregard them.

In the original manuscript, we discussed the influences of structural changes on the band gaps in the Supplementary Information, Section 3: Details of DFT calculations, on pages 3 and 4. We have added this to the discussion on page 4/5 of the SI:

Mostly importantly, our fitting procedure only makes use of the shape and relative positions of main peaks (Figure 2 in main manuscript). These remain almost unchanged. So we can disregard the differences introduced by the small changes in lattice parameters.

3. There is a typo in the numerator of the goldschmidt tolerance factor in both the main manuscript and SI: r_m should be r_x . It looks like the values were calculated correctly though.
→ We thank the reviewer for spotting the mistake and we have corrected it in both the main manuscript and in the SI.

4. While there is no extensive study of the band alignment of all of the Sn and Pb halide perovskite materials there have been several of MAPbI₃ (most of which the authors cite), but it could be helpful if the authors compare their values for the systems that have been done and put them in context. For instance: <http://dx.doi.org/10.1016/j.solener.2015.07.040>
→ The reviewer is right that so far we neglected to put our values into proper context here. Giving a broad overview over all published values and the reasons why they scatter so much is however rather a topic for a review paper and beyond the possibility of the limited space given here. We have now included the following section on page 6 to give the reader an idea of the variations found in literature:

Most of these materials have not been reported before, so we cannot in general discuss comparability of the values found here to previous reports. Even for the more extensively studied systems, published values usually scatter quite significantly due to issues in film preparation and data evaluation, as already mentioned in the introduction. For example, for MAPbI₃ values between 5.1 eV and 6.65 eV have been published³⁴ and for MASnI₃ a similarly broad distribution of 4.73 to 5.47 eV is found⁴⁷⁻⁵⁰. This emphasizes once more the need for a consistent study with an unambiguous data evaluation process.

5. Perhaps also a discussion how accurate the absolute values of the IE and EA are compared to the relative values are.

- The accuracy of absolute values extracted from the experiment is mostly influenced by the reproducibility of the materials' fabrication. The variation in different samples result in the error bars given in Table 1, which are in the range of ± 0.1 eV. Furthermore, the experimental broadening (introduced by the excitation source, sample temperature, detector response function, etc.), as well as inherent factors (for example surface sensitivity) have an effect on the accuracy of the PES measurements. These two factors however play no role in our data analysis, since we determine the band onset positions from the DFT calculated spectra. The only issues here could arise from not correctly calculating the DFT spectra, or by misaligning the measured and the DFT-calculated DOS. The former is discussed in our reply to point 2 of reviewer#2, while the latter can be done with an accuracy of a few 10 meV (which is much lower than the sample-to-sample variation). Since we already have a high confidence regarding the extracted absolute values, there is no reason why the relative values, comparing different perovskite materials, should be higher than that. In order to refer the readers to the error bars, we added on page 10 the following sentence to the discussion of the general energy level trends:

The values are taken from Table 1, where the absolute errors of the respective measurements can be found.

REVIEWERS' COMMENTS:

Reviewer #1 (Remarks to the Author):

The revised manuscript has resolved all my concerns in my original report. The paper can be accepted as is, and I expect this work will have a broad impact on the field.

Reviewer #2 (Remarks to the Author):

The authors have done a fair job at addressing both reviewer concerns. While my overall recommendation is to publish without further review, I would encourage the authors to add some of what they addressed in their response into the SI of the manuscript. Although not stemming from my questions, I found both the discussion on change of IE with precursor ratio and the comparison of the the IE and EA values using different approaches (log, linear, and the method used in the manuscript) very relevant to the field. In the end the information in this paper is meant to help with device design. Information from the IE vs precursor ratio gives valuable insight into which materials are more sensitive (something that is good to know). As far as comparing the 3 methods in the authors response they state that "some of the perovskites are known to be very defect tolerant, others are not, and in those cases we observe a strong variation in log scale band onsets." Here again I would say this is very valuable insight for the community (information and a technique to quantify a materials sensitivity to defects, something that will also be important for eventual device development).

I do understand that both of these aspects are quite technical with many readers not being interested. However I think they can both be incorporated with very minor reference in the main manuscript (for the "general audience") and actually discussed in the SI where an expert could delve in, and I encourage the authors to do as such. That being said, I do think the manuscript is publishable as is and leave it up to the authors to make the final decision. Whatever is decided no further review is needed

Response Letter

We thank the reviewers for evaluating the revised version of our manuscript “Absolute energy level positions in tin and lead based halide perovskites”. Below is the point-by-point response to the reviewer’s comments.

Reviewer #1 (Remarks to the Author):

The revised manuscript has resolved all my concerns in my original report. The paper can be accepted as is, and I expect this work will have a broad impact on the field.

We thank the reviewer for his/her positive evaluation and for once more stating the broad impact this work will have on the research community.

Reviewer #2 (Remarks to the Author):

The authors have done a fair job at addressing both reviewer concerns. While my overall recommendation is to publish without further review, I would encourage the authors to add some of what they addressed in their response into the SI of the manuscript. Although not stemming from my questions, I found both the discussion on change of IE with precursor ratio and the comparison of the the IE and EA values using different approaches (log, linear, and the method used in the manuscript) very relevant to the field. In the end the information in this paper is meant to help with device design.

Information from the IE vs precursor ratio gives valuable insight into which materials are more sensitive (something that is good to know). As far as comparing the 3 methods in the authors response they state that "some of the perovskites are known to be very defect tolerant, others are not, and in those cases we observe a strong variation in log scale band onsets." Here again I would say this is very valuable insight for the community (information and a technique to quantify a materials sensitivity to defects, something that will also be important for eventual device development).

I do understand that both of these aspects are quite technical with many readers not being interested. However I think they can both be incorporated with very minor reference in the main manuscript (for the "general audience") and actually discussed in the SI where an expert could delve in, and I encourage the authors to do as such. That being said, I do think the manuscript is publishable as is and leave it up to the authors to make the final decision. Whatever is decided no further review is needed

We thank the reviewer for once more carefully assessing the revised version and appreciate the time spent looking through the additional data, which was originally prepared for Reviewer 1. We are grateful for the recommendation of a publication.

The reviewer raises the point that the two topics discussed in the revision process, namely the off-stoichiometry sample variations and the issue of lin/log onset readout, are relevant enough to include in the Supplementary Information. After careful consideration we now indeed included one of the graphs

with the stoichiometry variation into the Supplementary information, as Supplementary Figure 10 and shortly discuss there that some samples show stronger variations than others. In the main manuscript we added on page 9:

It is often reported that variations in processing can lead to sample-to-sample variation, either due to variations in film stoichiometry or partial or complete transition into different crystal structures. For some examples and a more detailed discussion of these effects we refer to the Supplementary Figures 7 to 10.

Regarding the second topic of lin/log readout procedure: as indicated already in the first revision round this would result in quite an extensive discussion, first of the disadvantages and advantages of the different approaches, and another set of Figures (18 graphs) would have to be included in the Supplementary. Already now this document contains a lot of information, all of which we consider of interest to a wide range of readers. Adding another lengthy discussion that only interests a specific subgroup of researches would take the focus away from the main results of this work. Since the reviewer left the final decision of what to include in the Supplementary to the authors, we decided in the end to not include this second part.